# An objective collapse model without state dependent stochasticity

Lotte Mertens,[1, 2] Matthijs Wesseling,[1] and Jasper van Wezel[1]

[1]*Institute for Theoretical Physics Amsterdam, University of Amsterdam,*
*Science Park 904, 1098 XH Amsterdam, The Netherlands*
[2]*Institute for Theoretical Solid State Physics, IFW Dresden, Helmholtzstr. 20, 01069 Dresden, Germany*

(Dated: March 16, 2023)

The impossibility of describing measurement in quantum mechanics while using a quantum mechanical model for the measurement machine, remains one of its central problems. Objective collapse theories attempt to resolve this problem by proposing alterations to Schrödinger's equation. Here, we present a minimal model for an objective collapse theory that, in contrast to previous proposals, does not employ state dependent stochastic terms in its construction. It is an explicit proof of principle that it is possible for Born's rule to emerge from a stochastic evolution in which no properties of the stochastic process depend on the state being evolved. We propose the presented model as a basis from which more realistic objective collapse theories can be constructed.

## I. INTRODUCTION

Quantum physics gives an extremely accurate description for the dynamics of systems consisting of up to at least $10^6$ atoms [1, 2]. Contrarily, in the macroscopic realm of objects comprised of $10^{18}$ atoms or more, it has been experimentally verified that classical physics suffices to predict their collective dynamics [1, 3–11]. There is an inherent conflict between these two limits which becomes apparent when considering the measurement of microscopic particles with a macroscopic measurement machine that itself is built from quantum constituents [12–15]. This inconsistency is known as the quantum measurement problem and it remains the focus of an active field of research to this day [1, 16–20].

Proposals for solving the quantum measurement problem can be divided into roughly two classes: interpretations and objective collapse theories. The interpretations assume Schrödingers equation holds unaltered at all scales. The interpretations attempt to interpret the mathematical objects appearing in Schrödinger's equation in a way that explains why we always perceive only a single classical state for any macroscopic object, even if its wave function is superposed over many such states [1, 21, 22]. On the other hand, objective collapse theories assume that in between the microscopic and macroscopic worlds, at the so-called mesoscopic scale, a transition takes place from quantum to classical dynamics [16, 23, 24]. This transition is described by a small alteration of Schrödinger's equation that has no measurable effect on microscopic objects, but begins to dominate the dynamics in the mesoscopic regime [1, 21, 22]. The result is a reduction or 'collapse' of quantum superpositions into a basis of classical states, which becomes instantaneous in the macroscopic limit. State of the art experimental efforts have recently begun to probe the mesoscopic region where deviations from Schrödinger's equation predicted by objective collapse theories may become observable [1, 3–11]. These efforts attempt to create superpositions and observe quantum interference on mesoscopic scales, and typically involve objects of mesoscopic size, such as a nano-mechanical resonator or a large molecule being superposed over different vibrational modes or positions.

The precise dynamics predicted by objective collapse theories in the mesoscopic realm depend on the details of the particular theory being considered. In order to reproduce the observed properties of the collapse process in the macroscopic limit however, any objective collapse theory needs to at least contain stochastic, non-unitary [17], and non-linear[25] elements in its time evolution. Together, these need to ensure that the collapse dynamics give rise to Born's rule when averaged over many individual collapse processes [22, 26]. In existing objective collapse theories, this outcome is explicitly imposed in the proposed dynamical equations by multiplying the stochastic variable with a non-linear, state dependent factor [1, 16, 23, 24, 27–31]. The origin of this coupling remains unclear [1]. The resulting probability distribution function of the combined non-linear stochastic term inevitably depends on the state it acts on, which necessarily implies the stochastic term has complete information on the state. Here, we show that this feature can be avoided by explicitly constructing an objective collapse theory in which the non-linear elements are separate from any stochastic term. The stochastic term may still depends on state-independent properties of the system (such as the coupling constant defining magnetic interactions between spins in a magnet), but *not* on properties of the state the system is in (such as the weights of particular spin configurations in the wave function of a magnet).

Similar to conventional objective collapse theories, a relation between model parameters is required to obtain Born's rule. However, the model presented here is distinguished from existing collapse models because Born's rule emerges without a state-dependent probability distribution function for the stochastic variable. This proof of principle is worked out in the context of a Heisenberg antiferromagnet acting as a measurement machine for staggered magnetisation, and we demonstrate that it exhibits all characteristics required of an objective collapse model. The proposed theory is not intended to be a realistic proposal for objective collapse in antiferromag-

netic materials, but rather serves as a minimal example that can be used as the basis for constructing more general and more realistic objective collapse models.

The article is organised as follows. First, the requirements that should be met by any objective collapse model are summarised. Next, a recipe is given for constructing an objective collapse theory applying to a general two-state measurement. Each step is then applied to the specific situation of a Heisenberg antiferromagnet, and the properties of the ensuing collapse process are detailed. Finally, an interpretation is given for the proposed dynamics, and the advantages of the proposed model over existing theories are discussed.

## II. REQUIREMENTS FOR ANY OBJECTIVE COLLAPSE THEORY

Regardless of their detailed constructions, all objective collapse theories propose an alteration or correction to Schrödinger's equation that can be written as [32]:

$$i\hbar \frac{\partial}{\partial t} |\psi(t)\rangle = (H + \epsilon G) |\psi(t)\rangle \qquad (1)$$

Here, we (arbitrarily) use the Schrödinger picture, and explicitly include the strength $\epsilon$ of the term $G$ generating collapse.

The first requirement that any objective collapse model written in the form of Eq. (1) should obey, is that the time evolution it generates should reduce to standard quantum mechanics when acting on microscopic objects. This is ensured by assuming $\epsilon$ to be suitably small, so that any deviations from the usual quantum dynamics become apparent only at very late times scaling with $1/\epsilon$ (which could even lie beyond the current age of the universe). Despite the smallness of $\epsilon$, however, any successful collapse theory should also predict macroscopic objects that are somehow forced into a superposition of classically distinguishable states (f.e. by instantaneously coupling them to a microscopic quantum system), to very quickly collapse towards just one such state [33, 34]. This happens when the operator $G$ scales with some measure $N$ of the system size, such as the number of particles it involves, its mass, its volume, or the value of its classical order parameter, which distinguishes different classical states [1, 35]. The collapse time scale $\tau_c \propto 1/(N\epsilon)$ can then become arbitrarily small in the thermodynamic limit, while remaining arbitrarily large for microscopic objects [35].

With this requirement satisfied, the time evolution implied by Eq. (1) can be used to describe quantum measurement by splitting it into two stages [22]. First, a microscopic object is coupled to a macroscopic measurement device in such a way that part of the device, traditionally referred to as the 'pointer', becomes entangled state with the microscopic object. This is often called pre-measurement. It is then in a superposition of classically distinguishable states, or pointer states [36]. Sec-

ondly, this superposition will almost instantaneously reduce to just a single pointer state, after which the measurement outcome can be read off. The pre-measurement is dominated by the coupling between the measurement device and the microscopic object, which is encoded in $H$, while $G$ induces the latter part of the collapse to a single pointer state.

This description of measurement brings to the fore a second requirement for any objective collapse theory: it should cause stable quantum state reduction. That is, when acting on a macroscopic system, $G$ should cause it to be reduced to a single pointer state. Moreover, once the measurement device is localised in a single pointer state, it should not be able to spontaneously evolve out of it. The requirement of reduction to a single pointer state turns out to be readily fulfilled for example by realising that the dynamics generated by the usual Schrödinger equation (with $G = 0$) is unstable [17]. That is, if $G$ is not Hermitian and couples to an order parameter of any symmetry-breaking system (such as the position of a crystal or the magnetization of a magnet), it has been shown that even an infinitesimal value for $\epsilon$ suffices to instantaneously collapse sufficiently large macroscopic systems into a pointer basis [17].

Notice that although $G$ being non-Hermitian implies non-unitary time evolution, this does not present any fundamental problems with regard to wave function normalisation. Upon redefining expectation values as $\langle O \rangle \equiv \langle \psi | O | \psi \rangle / \langle \psi | \psi \rangle$, all of the physical predictions of quantum mechanics can be recovered even with a time-dependent norm [17, 25].

The possible lack of energy conservation under non-unitary time evolution similarly does not pose a problem [37]. The conservation of $\langle H \rangle$ is ensured in the thermodynamic limit if the non-Hermitian field couples (with non-vanishing strength) only to the order parameter, such as the centre of mass position or magnetization, and not to any internal degrees of freedom such as sound or spin wave excitations. This way, the non-Hermitian field causes transitions only between states that are degenerate in the thermodynamic limit, and hence have no effect on the total energy [17]. For mesoscopic systems, small fluctuations in $\langle H \rangle$ provide one possible way to experimentally distinguish the predictions of objective collapse theories from those of unitary interpretations. [38].

Finally, the third requirement on the predicted dynamics of the pointer state in any objective collapse theory, is that it should reproduce Born's rule. That is, the relative frequency of any particular measurement outcome should equal the squared weight of the corresponding component in the initial pointer state wave function. This requirement also implies that in general, a given initial state needs to be able to collapse to different pointer states representing different measurement outcomes. This variation in possible end states necessarily implies the presence of a stochastic variable in the time evolution generator, which changes value from one collapse process to the next, and possibly even within a single process. It

has recently been shown that besides a stochastic component, it is also necessary for the generator to contain a non-linear component (independent of normalisation) to be able to satisfy Born's rule [25].

In summary, the ingredients necessary for any objective collapse theory are that its time evolution operator should be non-unitary, should scale with the system size, should couple only to the order parameter of the system in the thermodynamic limit, and should contain both non-linear and stochastic terms.

## III. CONSTRUCTING AN OBJECTIVE COLLAPSE THEORY

Here, we explicitly construct an objective collapse theory by systematically including all necessary ingredients identified above. We will construct the theory in the context of an antiferromagnet acting as a measurement device [35], but the procedure is readily generalised. The result differs in a crucial aspect from the many existing flavours of objective collapse theory [15, 23, 24, 27, 28]: it gives rise to Born's rule without the stochastic contribution to the evolution having any knowledge of the state of the system. This way, Born's rule is not imposed on the dynamics by the way the theory is formulated, but rather emerges spontaneously in the thermodynamic limit.

### A. Introducing the pointer basis

Consider a macroscopic system governed by the Hamiltonian $H$, consisting of many internal degrees of freedom. If the system represents a measurement device, it will be described by a classically accessible collective variable describing properties of the system as a whole, such as its centre of mass or total magnetization, which we call the 'pointer' [17, 35]. The possible states for the pointer correspond to different symmetry-broken configurations of the system. The pointer states could be different positions along a dial of an actual pointer, they could be different configurations of text on a computer screen displaying measurement outcomes, or they could be any other set of classically distinguishable configurations. To be specific, we will here consider an antiferromagnet whose pointer states consist of different orientations of the spins making up the antiferromagnet.

In fact, it suffices to consider only initial states of the antiferromagnet superposed over two states, with opposite staggered magnetisation. If the objective collapse theory does not reproduce all aspects of measurement for two-state superpositions, it follows by induction that it can also not work for any more complicated initial state. Formulating collapse dynamics for two-state superpositions thus provides a minimal model for an objective collapse theory. Moreover, we will focus only on the pointer states of the antiferromagnet and ignore all of its internal degrees of freedom because these are the only degrees of freedom involved in spontaneous symmetry breaking [25].

To be specific, we consider a Heisenberg antiferromagnet, with positive isotropic coupling $J$ between neighbouring quantum spins of size $1/2$:

$$H^{\mathrm{AF}} = \sum_{j=1}^{N} J\, \mathbf{S}_j \cdot \mathbf{S}_{j+1}.$$

The spins on neighbouring sites will anti-align, forming two sub-lattices $A$ (even sites) and $B$ (odd sites) with opposite spins. Upon taking the Fourier transform it becomes clear that the collective parts of the Hamiltonian $H_0$ (with $k = 0, \pi$) decouple from the internal spin wave degrees of freedom (with $k \neq 0, \pi$) [39]:

$$H_0 = H^{\mathrm{AF}} - H^{\mathrm{internal}} = \frac{4J}{N} \mathbf{S}_A \cdot \mathbf{S}_B. \tag{2}$$

$H_0$ is the so-called Lieb Mattis Hamiltonian [40].

Notice that in the thermodynamic limit, all states with different values of the total spin $\mathbf{S} = \mathbf{S}_A + \mathbf{S}_B$ become degenerate with the ground state, while maintaining only a vanishing contribution to the free energy of the antiferromagnet [39].

The usual theory of spontaneous symmetry breaking (SSB) describes the emergence of stable states that are not invariant under a symmetry of the Hamiltonian. For the Lieb-Mattis Hamiltonian of Eq. (2) the spin-rotational symmetry of the antiferromagnet can be broken by adding an infinitesimal symmetry breaking field to the Hamiltonian:

$$H = H_0 - B(S_A^z - S_B^z). \tag{3}$$

depending on the sign of $B$, it represents a magnetic field that points either up or down along the $z$-direction on the $A$-sublattice, while it points in the opposite direction on the $B$-sublattice. The pointer state with staggered magnetisation $S_A^z - S_B^z = \pm N/2$ is singled out in the thermodynamic limit as the unique ground state by even a vanishingly small field $B = \pm|B|$ [39]. These states are stable in the limit of vanishing $B$ despite the fact that they break the spin-rotational symmetry of $H_0$. They are the pointer states of the antiferromagnetic measurement machine considered here.

### B. Introducing non-unitary dynamics

Time evolution generated by the Schrödinger equation is necessarily invertible, but as claimed before [17], it is also unstable to non-unitary perturbations. Applying the same principles of equilibrium spontaneous symmetry breaking, but now in the setting of non-equilibrium time evolution then generates the spontaneous breakdown of invertible time evolution in the presence of even a vanishingly small non-Hermitian term.

We will explicitly consider a non-Hermitian version of the symmetry-breaking perturbation given by writing $i\epsilon B(S_A^z - S_B^z)$, with $\epsilon$ vanishingly small and $B$ finite. In addition, we break invertible time evolution on the level of the collective Hamiltonian by introducing the infinitesimal Wick rotation: $H_0 \to (1 + \epsilon i)H_0$. This perturbation can be interpreted as the leading order contribution arising from a full-fledged non-unitary theory for physics beyond Schrödinger's equation, and we neglect all higher order contributions.

Both of the sublattice spin operators $\mathbf{S}_{A/B}$ appearing in the collective Hamiltonian scale with the system size $N$. The total strength of the non-unitary contribution to the pointer dynamics therefore scales with $N\epsilon$, and displays the non-commuting limits typical of spontaneous symmetry breaking [39]:

$$\lim_{\epsilon \to 0} \lim_{N \to \infty} N\epsilon = \infty$$
$$\lim_{N \to \infty} \lim_{\epsilon \to 0} N\epsilon = 0. \tag{4}$$

This implies that the speed of the collapse process induced by the non-unitary term depends on the size of the pointer. If $N$ is small, the non-unitary part is negligible for any sufficiently small $\epsilon$. In the thermodynamic limit however, the number of spins making up the collective anti-ferromagnetic state of the pointer is so large that any infinitesimal coupling suffices to qualitatively influence the pointer dynamics [17, 35].

### C. Introducing an external stochastic field

Once an object in the thermodynamic limit is in a state with a single, definite value for the order parameter, it does not unitarily evolve out of that state within any measurable time, even if the direction of the symmetry-breaking field changes. There is therefore no reason to constrain $B$ to be a static external field, and we will allow it to vary in time. In fact, since it represents an infinitesimal symmetry breaking field originating from sources beyond the control of any feasible experiment or observation, we assume the direction of $\mathbf{B}(t)$ to vary randomly in time, with correlation time $\tau_r$. At the start of a measurement process, the direction of the symmetry breaking field is then fully random and has equal probability of being aligned with any particular pointer state.

Notice that at this point we have two different non-Hermitian contributions to the Hamiltonian. The first originated from Wick rotating the original symmetric Hamiltonian, and yields spontaneous non-unitary dynamics for the system even without any external influences. The second term is a non-Hermitian version of a symmetry breaking field. This is due to the interaction with an (as yet undefined) external actor and introduces a stochastic contribution to the evolution. The interplay between the two non-Hermitian terms will determine the collapse dynamics and measurement outcomes.

### D. Introducing non-linearity

In order to reproduce Born's rule, the time evolution operator necessarily needs to contain a non-linear term [25]. As argued before, the non-linearity needs to go beyond normalization of the wave function, as that is already made redundant by the definition $\langle O \rangle \equiv \langle \psi | O | \psi \rangle / \langle \psi | \psi \rangle$. One way to introduce such a non-linearity is by considering in the Hamiltonian only the effective, collective influence of all microscopic degrees of freedom on the order parameter. For general symmetry-breaking systems taken out of equilibrium it is well-known that the Gross-Pitavskii equations (a form of dynamical mean field theory) give a good approximation of the dynamics of the order parameter [41–45].

In our case, the Gross-Pitaevskii approach consists of approximating the interaction between sub-lattices by each spin being exposed to the magnetic field generated by the spins in the opposing sub-lattice: $\mathbf{S}_A \cdot \mathbf{S}_B \approx \langle \mathbf{S}_A \rangle \cdot \mathbf{S}_B + \mathbf{S}_A \cdot \langle \mathbf{S}_B \rangle$. These terms can be evaluated at each time step, giving a set of differential equations that define the time evolution starting from a given initial state.

The Gross-Pitaevskii approach yields a good approximation for the collective dynamical response seen in isolated systems without ensemble averaging over multiple experiments [43–45]. Thus the approximate form of the self-interaction, containing factors like $\langle \psi | \mathbf{S}_A | \psi \rangle$, which have the appearance of expectation values, actually do not imply ensemble averages or measurements being performed. They can be considered as the instantaneous collective magnetisation of all spins in one sub-lattice influencing the spins in the other within a *single* experiment.

### E. Assembling the objective collapse theory

Putting everything together, the time evolution of equation (1) for the antiferromagnetic pointer state can be written as:

$$i\hbar \frac{\partial}{\partial t} |\psi(t)\rangle = \left[ \frac{4J}{N}(1 + i\epsilon)\left( \frac{\langle \psi(t)| \mathbf{S}_A |\psi(t)\rangle}{\langle \psi(t)|\psi(t)\rangle} \cdot \mathbf{S}_B + \frac{\langle \psi(t)| \mathbf{S}_B |\psi(t)\rangle}{\langle \psi(t)|\psi(t)\rangle} \cdot \mathbf{S}_A \right) + i\epsilon \mathbf{B}(t) \cdot (\mathbf{S}_A - \mathbf{S}_B) \right] |\psi(t)\rangle \tag{5}$$

As noted before, it is a necessary requirement for any objective collapse theory to describe the quantum state reduction of a measurement machine that is initially superposed over just two pointer states. We therefore consider the simplest possible case of a state composed of two eigenstates of the order parameter:

$$|\psi(t)\rangle = n(t)e^{i\xi(t)/2} \left( e^{i\phi(t)/2} \cos(\theta(t)/2) |\uparrow\downarrow\rangle \right.$$
$$\left. + e^{-i\phi(t)/2} \sin(\theta(t)/2) |\downarrow\uparrow\rangle \right). \quad (6)$$

Here, $n(t)$ represents the norm of the wave function, $\xi(t)$ its overall phase, $\phi(t)$ the relative phase between pointer states, and $\theta$ is the angle determining their relative weights. The states $|\uparrow\downarrow\rangle$ and $|\downarrow\uparrow\rangle$ are arbitrarily chosen to lie along the $z$-axis.

The direction of the symmetry-breaking field $\mathbf{B}(t)$ in equation (5) does not necessarily align with direction of the collective staggered magnetisations in the state $|\psi(t)\rangle$. Because of the rigidity associated with the spontaneously broken symmetry of the antiferromagnetic state, however, any components of the field orthogonal to the magnetisation direction will take a time proportional to the system size to have any significant effect [39]. Without loss of generality, we can therefore approximate the interaction terms as $\mathbf{B} \cdot \mathbf{S}_{A,B} \approx B_0 \cos(\chi) S_{A,B}^z$, with $B_0$ the amplitude of the symmetry-breaking field, and $\chi$ the angle between $\mathbf{B}$ and the $z$-axis. Choosing a random direction for the three-dimensional vector $B$ then corresponds to randomly sampling a value for $\chi$ from the probability density function $f = \sin(\chi)/2$. Following the same reasoning, we can also approximate the Gross-Pitaevskii terms by their projections onto the $z$-axis: $\langle \mathbf{S}_{A,B} \rangle \cdot \mathbf{S}_{B,A} \approx S_{A,B}^z S_{B,A}^z$.

Inserting all definitions into the Hamiltonian and calculating the time dependence of $\theta$ from equation (5) as described in appendix A finally yields:

$$\dot{\theta} = -\frac{JN\epsilon}{\hbar} \sin(\theta) \left( \cos(\theta) - \frac{B_0}{J} \cos(\chi(t)) \right). \quad (7)$$

This equation shows the dynamics of the relative weights parameterised by $\theta(t)$ to be independent of the variables $n(t)$, $\xi(t)$, and $\phi(t)$ so that we only need to consider the relative weights when modelling the objective collapse process. In particular, it justifies the claim that the overall norm $n(t)$ can be safely absorbed into a redefinition of the expectation value without affecting any of the collapse dynamics.

## IV. COLLAPSE DYNAMICS

Having formulated a model for the quantum state reduction of a single superposition of pointer states, we will discuss the collapse dynamics it gives rise to. We will determine the outcomes of the predicted evolution for different regimes of pointer size, the frequencies with which particular outcomes are obtained, and we will check that the model meets the minimal requirements for objective collapse theories formulated above.

## A. Microscopic region

In the microscopic limit, the pointer itself is a quantum system that typically consists of a small number of constituent particles (roughly, fewer than $10^6$ atoms [1]). Taking the strength $\epsilon$ of the non-unitary perturbation to be infinitesimally small such that $N\epsilon \to 0$, then renders the entire dynamics described by equation (7) negligible on any measurable time scale. As expected, only the regular, unitary quantum dynamics governed by Schrödinger's equation remains in this limit.

## B. Macroscopic region

To describe a macroscopic pointer we consider the thermodynamic limit $N \to \infty$. The collapse dynamics of equation (7) then dominates the time evolution for any non-zero value of $\epsilon$, thus satisfying the first requirement for objective collapse theories. The collapse time $\tau_c$ in this limit will be far shorter than any correlation time of the random variable $\chi(t)$. The angle $\chi$ is therefore approximately constant during the collapse process, and the evolution of $\theta(t)$ can be depicted in a flow diagram like the one in figure 1. Here, $\dot{\theta}$ is depicted as a function of the instantaneous value of $\theta$ itself, for different values of the random variable $\chi$ ranging from zero in black (top curve) to $\pi$ in orange (bottom). The qualitative properties of the dynamics can be directly seen in the plot. For positive values of $\dot{\theta}$ the angle $\theta$ will increase over time. Since the black (top) line for $\chi = 0$ lies entirely above the $\dot{\theta} = 0$ axis, the value of $\theta$ will increase over time regardless of its initial value. The state will thus flow towards $|\downarrow\uparrow\rangle$ and stay there forever. The flow is in the opposite direction for $\chi = \pi$, while for intermediate values of $\chi$ the outcome of the collapse dynamics depends on the initial value of $\theta$.

The value of $\chi$ determines the initial value of $\dot{\theta}$, and its sign will not change during time evolution, thus driving the state towards either of the two components in the initial superposition, where it remains forever after. This is the manifestation of the stability of quantum state reduction in the present model, which was the second requirement for objective collapse theories. We show in subsection IV C that variations in the value of $\chi$ after the state has collapsed do not affect this stability, and the macroscopic antiferromagnet will not spontaneously evolve into a superposition of pointer states.

To determine the probability of finding the final state $|\uparrow\downarrow\rangle$ as the outcome of an individual collapse process, we integrate the probability density for finding a particular value of $\chi$ over all values that lead to the desired measurement outcome. Since the function $\sin(\theta)$ is always positive for $\theta \in [0, \pi]$, the sign of $\dot{\theta}$, and thus the result of

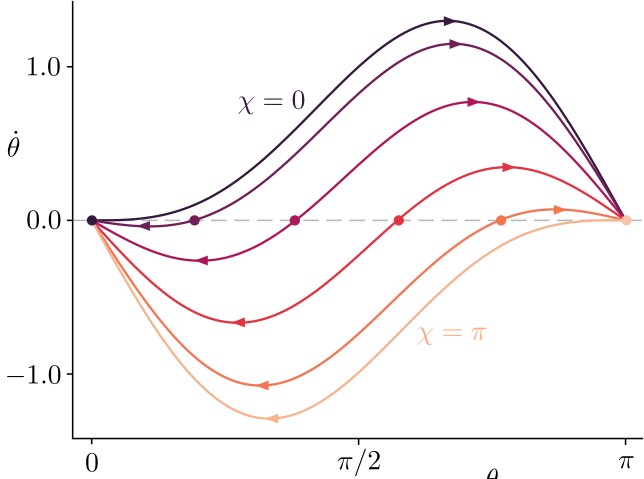

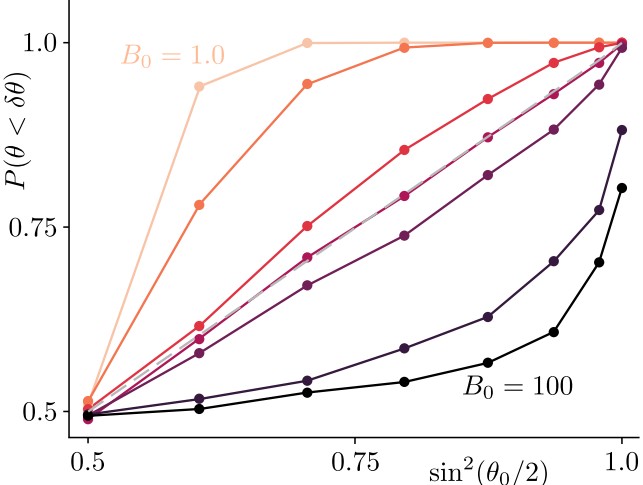

FIG. 1. **Flow diagram for the pointer state dynamics.** The rate of change $\dot{\theta}$ is depicted as a function of the instantaneous angle $\theta$ for different values of the random variable $\chi$, ranging from $\chi = 0$ (dark, top) to $\chi = \pi$ (light, bottom). Here, we used $J = B_0$. The arrows show the direction of flow, while the dots indicate fixed points.

FIG. 2. **Reproducing Born's rule.** The relative frequency of evolutions coming within $\delta\theta$ of the state $|0\rangle$ is plotted as a function of the initial weight $\sin^2(\theta_0/2)$. Dots represent calculated values, while lines are included as a guide to the eye only. The diagonal dashed line indicates Born's rule, while the differently coloured data sets show different values of the parameter $B_0$, ranging from $B_0 = 1.0$ at the top through 2.0, 5.0, 7.0, 10, 50, and $B_0 = 100$ at the bottom. We set $J = 1$ throughout. For each set of parameter values, $10^4$ instances of the dynamics are calculated for a maximum of $20^3$ time steps. Taking $\tau_r$ small compared to $\tau_c$ and the size of time steps, the random variable $\chi \in [0, \pi]$ is sampled each time step from the probability density function $\sin(\chi)/2$, which yields a flat distribution for the value of $\cos(\chi)$.

the collapse dynamics, is determined entirely by the sign of $\cos(\theta(0)) - (B_0/J)\cos(\chi)$. The probability for finding the final state $|\uparrow\downarrow\rangle$ is thus given by:

$$P_{|\uparrow\downarrow\rangle} = \int_0^\pi \frac{\sin(\chi)}{2} \Theta\left(\cos(\theta_0) - \frac{B_0}{J}\cos(\chi)\right) d\chi$$

$$= \int_{\arccos\left(\frac{J}{B_0}\cos(\theta_0)\right)}^\pi \frac{\sin(\chi)}{2} d\chi$$

$$= \frac{1}{2}\left(1 + \frac{J}{B_0}\cos(\theta_0)\right) \qquad (8)$$

Here $\Theta(x)$ is the Heaviside step function, and we defined $\theta(0) = \theta_0$. In the second line the step function is converted into a lower bound on the integral. The result in the final line equals $\cos^2(\theta_0/2)$ if and only if $J = B_0$. This would correspond to Born's rule, as can be read of from Eq. 6, and thus the third requirement on objective collapse theories being satisfied. We discuss possible physical mechanisms that may result in such a relation between $J$ and $B_0$ in section V

### C. Mesoscopic region

For mesoscopic systems, in which the number of constituent particles is neither small nor approaching the thermodynamic limit, there will be a range of system sizes such that the collapse time $\tau_c$ is larger than the correlation time $\tau_r$ of the random variable but still finite. The value of $\chi$ will then vary in time during the collapse process.

The probability of obtaining a given measurement outcome in this regime can in principle be written as a

path integral in a straightforward generalisation of equation (8). Here, we take the complementary approach of evaluating the probabilities numerically by sampling random values for $\chi(t)$ from a distribution that enforces both its correlation time $\tau_r$ and its long-time uniform probability density profile [46, 47].

Taking very large values for the ratio $B_0/J$ while keeping $\tau_r/\tau_c$ finite, the stochastic term in equation (7) dominates the dynamics and the measurement outcome is determined primarily by the sign of $\cos(\chi(t))$ at early times. Since $\cos(\chi(t))$ starts from a flat distribution, the probability of finding either pointer state is one half, independent of the initial state.

In the opposite extreme of very small $B_0/J$ the stochastic term becomes negligible and the collapse statistics is determined entirely by the initial sign of $\cos(\theta)$. The probability for finding the measurement outcome $|\uparrow\downarrow\rangle$ is then one for $\theta < \pi/2$, and zero otherwise.

For intermediate values of the ratio $B_0/J$ the collapse statistics may be expected to smoothly interpolate between these extreme behaviours, suggesting the possibility of reproducing Born's rule for fine-tuned parameter values, as shown in figure 2 for the example of small $\tau_r/\tau_c$. Indeed, we numerically established that for any value of the parameters $N$, $J$, and $\tau_r$ in equation (7), there is a value of $B_0$ which results in Born's rule being obeyed by

the collapse statistics. For the objective collapse model of equation (7) the parameters yielding Born's rule in the mesoscopic regime are found to approximately follow the relation:

$$J \approx 2B_0^2 N \tau_r \qquad (9)$$

Notice that for the macroscopic regime with $\tau_r > \tau_c$, we saw before that Born's rule is obeyed if $J = B_0$. At the point $\tau_r = \tau_c = 1/(2JN)$ connecting the two regimes, the mesoscopic relation of equation (9) connects continuously to the macroscopic one.

Notice that a correlation between parameters is not an assumption of Born's rule, but rather an indication of an underlying physical relation between the parameters. A example of similar correlations occurs in Einstein's use of fluctuation-dissipation to explain why the diffusion and drift terms in Brownian motion must be fine-tuned with respect to one another [48, 49]. The fine-tuning seen here similarly signals the fact that the physical observables represented by these parameters are related.

Given the fine-tuned relation between model parameters, equation (7) is an objective collapse model that reproduces Born's rule, assuming that the final states obtained are stable. That this is not obvious in the presence of a time-varying stochastic parameter is clear from the observation that such a term always destabilises the solutions of linear time evolution equations [25]. In the present case, however, the non-linear nature of the collapse dynamics protects the stability of the pointer states. This can be easily seen by noticing that $d\theta/dt$ near one of the poles of the Bloch sphere is directed towards the pole for almost all values of the random parameter.

### D. Lower bound on the correlation time

The small but non-zero chance of a state evolving away from the poles can be used to further constrain the model parameters. First, approximate the time-varying stochastic term as a process in which $\cos(\chi(t))$ is constant for intervals equal to the correlation time $\tau_r$ and chosen randomly from a flat distribution in every new interval. Starting from an equal-weight superposition, the state of the system then comes within roughly $\delta\theta \sim \exp(-\tau_r/\tau_c) \sim \exp(-JN\epsilon\tau_r)$ of a pole of the Bloch sphere within a single correlation time $\tau_r$. We can consider the collapse of an object of size $N$ stable if after collapsing into one of these regions, it may be expected to stay there for at least the age of the observable universe ($\tau_u \sim 4.4 \cdot 10^{17}$ s). The probability of finding a value for $\chi$ that would take the system out of the interval $[0, \delta\theta]$ is equal to $\sin^2(\delta\theta/2)$. Roughly, the collapse is stable within a time equal to the age of the universe if $\tau_r/\sin^2(\delta\theta/2) > \tau_u$, or equivalently if $2JN\epsilon\tau_r \gtrsim \ln(\tau_u/(4\tau_r))$. More generally, observing an object of size $N$ to remain collapsed for a time $\tau_u$ thus provides the value of $\epsilon\tau_r$ with a lower bound. This

allows existing methods constraining objective collapse theories [33] to be applied to the current model.

Further experimental predictions may be deduced from the dependence of the predicted collapse time on the system size. These can for example result in bounds on parameter values using experiments similar to those proposed in the context of other theories, which track quantum interference effects for superpositions of ever heavier or larger objects [50].

Similarly, further constraints on the model may be obtained by excluding the possibility of super-luminal communication [51, 52].

## V. DISCUSSION

The objective collapse model constructed here for an antiferromagnet superposed over two pointer states meets all minimal requirements for a theory of quantum measurement. It predicts a negligible effect on the dynamics of microscopic systems, which thus evolve purely according to Schrödinger's equations. At the same time, it predicts macroscopic systems to instantaneously collapse towards a single classical state. Moreover, owing to the combination of stochastic and non-linear ingredients, the classical end states are stable and the frequencies of outcomes obey Born's rule.

The construction of the model employed a series of generic steps that suggest the possibility of constructing similar objective collapse theories in more general settings. To wit, because the pointer was assumed to have a spontaneously broken symmetry, the order parameter dynamics separated from the effects of internal degrees of freedom and we could focus the model on only the collective properties of the antiferromagnet. The required stochastic nature of the collapse process is included in a natural way by assuming that the infinitesimal symmetry-breaking field is beyond the control of any experiment. Finally, its necessary non-unitary character is introduced in the form of an infinitesimal Wick rotation of the time axis, while the theory is rendered effectively non-linear by writing its dynamics in the form of Gross-Pitaevskii equations.

Although all of the steps above straightforwardly generalise to any symmetry-breaking pointer state and any initial state configuration, we find the fine-tuning of parameter values required for obtaining Born's rule not to be straightforwardly reproducible in more general settings. We therefore do not argue that the current model presents a realistic theory of nature. Rather, we present it as a proof of principle establishing that Born's rule can emerge from a consistent objective collapse theory without assuming it in the definition of the model.

This should be contrasted with existing objective collapse models, in which the probability density function governing the values attained by the stochastic component depend on the instantaneous state of the system. This is done either explicitly by using a probability den-

sity function that coincides with the instantaneous real-space wave function [28], or implicitly by considering a Wiener process whose probability density profile is multiplied by a state-dependent factor [1, 23, 27].

In the well-known QMUPL model for example [1, 53], all necessary ingredients for an objective collapse theory –a stochastic term, non-unitarity, and a non-linear term– are present. But the non-linear term multiplies the stochastic process and hence renders the probability density function of the stochastic variable dependent on the state of the system being measured. The stochastic term determines the probabilities of measurement outcomes, and the fact that its distribution scales precisely with the expectation value that one hopes to find as a result of the measurement dynamics amounts to introducing Born's rule in the definition of the model. Moreover, assuming the properties of the stochastic influence to depend on the object being measured severely complicates any physical interpretation for its origin.

In contrast, the model proposed here has separate non-linear and stochastic terms, which do not depend on one another in any way. The stochastic variables are therefore drawn from a state-independent probability density profile, and can be interpreted as originating from a universal, dynamically fluctuating, non-unitary noise field.

In conclusion, we established the possibility of constructing a model that satisfies all requirements of an objective collapse theory, without the presence of Born's rule being implied by a state-dependent stochastic term. We thus provide proof of principle for the possibility of Born's rule spontaneously emerging in quantum measurement, and suggest the presented model as a possible starting point in the search for further objective collapse theories that do not impose Born's rule.

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

## Appendix A: Time evolution on the Bloch sphere

The time evolution of quantum states can be written as:

$$\frac{\partial}{\partial t} |\psi(t)\rangle = \frac{i}{\hbar} \hat{F}(t) |\psi(t)\rangle \tag{A1}$$

with $F$ the (possibly non-Hermitian) time evolution generator. Specializing to the time evolution of the two-state superpostion of antiferromagnetic states discussed in the main text, we parameterize the state in its most general form as:

$$|\psi(t)\rangle = n(t)e^{i\xi(t)/2} \left( e^{i\phi(t)/2} \cos(\theta(t)/2) |\uparrow\downarrow\rangle + e^{-i\phi(t)/2} \sin(\theta(t)/2) |\downarrow\uparrow\rangle \right)$$

$$\Rightarrow \quad \frac{\partial}{\partial t} |\psi(t)\rangle = \frac{1}{2} \left( 2\dot{n}\cos(\theta/2) - n\dot{\theta}\sin(\theta/2) + i(\dot{\xi} + \dot{\phi})n\cos(\theta/2) \right) e^{i\xi/2} e^{i\phi/2} |\uparrow\downarrow\rangle$$

$$+ \frac{1}{2} \left( 2\dot{n}\sin(\theta/2) + n\dot{\theta}\cos(\theta/2) + i(\dot{\xi} - \dot{\phi})n\sin(\theta/2) \right) e^{i\xi/2} e^{-i\phi/2} |\downarrow\uparrow\rangle . \tag{A2}$$

Here, we introduced the time-dependent norm $n(t)$, overall phase $\xi(t)$, relative phase $\phi(t)$ and the angle $\theta(t)$ determining the relative weights of the wave function components. This expression gives the left hand side of equation A1. To find the right hand side, we recall the Hamiltonian for the antiferromagnet with non-unitary modifications to Schrödingers equation introduced in the main text:

$$\hat{H}|\psi(t)\rangle = \left[ \frac{4J}{N}(1+i\epsilon) \left( \frac{\langle\psi(t)| \hat{S}_A^z |\psi(t)\rangle}{\langle\psi(t)|\psi(t)\rangle} \hat{S}_B^z + \frac{\langle\psi(t)| \hat{S}_B^z |\psi(t)\rangle}{\langle\psi(t)|\psi(t)\rangle} \hat{S}_A^z \right) + i\epsilon B_0 \cos(\chi)(\hat{S}_A^z - \hat{S}_B^z) \right] |\psi(t)\rangle$$

$$= -\frac{N}{2} \left[ J(1+i\epsilon)\cos(\theta) - i\epsilon B_0 \cos(\chi) \right] ne^{i\xi/2} \left( e^{i\phi/2}\cos(\theta/2)|\uparrow\downarrow\rangle - e^{-i\phi/2}\sin(\theta/2)|\downarrow\uparrow\rangle \right), \tag{A3}$$

where in the second line we again focused on the two-state superposition of equation (A2), and wrote the non-linear contributions to the Hamiltonian as $\langle\psi(t)| \hat{S}_{A,B}^z |\psi(t)\rangle/\langle\psi(t)|\psi(t)\rangle = \pm N/4(\cos^2(\theta/2) - \sin^2(\theta/2)) = \pm N\cos(\theta)/4$. We also used the fact that each sublattice has $N/2$ spin halves to write $\hat{S}_{A,B}^z |\uparrow\downarrow\rangle = \pm N/4 |\uparrow\downarrow\rangle$.

Substituting equation (A2) for the left hand side and equation (A3) for the right hand side of (A1), we can separately equate the real and imaginary parts of the components for each of the two basis states. This gives four equations, which can be solved for the four parameters characterising the two-state superposition. The imaginary parts yield:

$$\dot{\xi} = 0$$
$$\dot{\phi} = -JN\cos(\theta)/\hbar. \tag{A4}$$

These are equal to the usual mean-field expressions for unitary time evolution a Lieb-Mattis antiferromagnet. The real parts of equation contain the non-unitary contributions and yield:

$$\frac{\dot{n}}{n} = \frac{JN\epsilon}{2\hbar} \cos(\theta) \left( \cos(\theta) - \frac{B_0}{J}\cos(\chi) \right)$$
$$\dot{\theta} = -\frac{JN\epsilon}{\hbar} \sin(\theta) \left( \cos(\theta) - \frac{B_0}{J}\cos(\chi) \right). \tag{A5}$$

The overall phase $\xi$ and normalisation $n$ do not appear in the equations for $\dot{\theta}$ and $\dot{\phi}$. This indicates that they remain unobservable even under the non-unitary time evolution, and can ignored when analysing the flow of $(\theta, \phi)$ on the Bloch sphere. Notice that if they could be observed there would immediately be super-luminal communication, explained in detail in [36, 54].