# Peer review of "An objective collapse model without state dependent stochasticity"

_SciPost Physics_

## Round 1 · Referee Report · Eric Aspling (Referee 1) · 2022-9-26

Strengths

1: Connection to experiment 2: Novel 3: Well-cited

Report

This is an excellent paper. The authors ideas are novel and compelling. The manuscript is well written and the citations are logical and plenty for a paper of this size.

I only $\textbf{suggest}$ the two changes listed below. However, the paper would be good with or without the suggestions.

1: Specify and elaborate specific experiments of interest to the authors. Although the citations are specific experiments, it would be helpful if the authors could summarize briefly how the experiment can shed light on their collapse theory. Even one single example so that the reader doesn't have to read and process multiple citations. One of the unique things about this method, is how connected perturbations of the Schrodinger equations are to experiment. I think slightly more emphasis on this would go a long way.

2: The second to last paragraph in section IV is quite remarkable, yet easily lost! I might suggest splitting this paragraph into two separate paragraphs or possibly another subsection, for any reader that is skimming. A graph of N vs. $\varepsilon \tau_r$ might be a really exciting way to see the relationship of system size and correlation time, and how these numbers actually stack up.

---

## Round 1 · Referee Report · Anonymous (Referee 2) · 2022-12-19

Strengths

None, I am afraid!

Weaknesses

  1. Incorrect criticism of objective collapse theories.

  2. Unsatisfactory claimed derivation of Born probability rule.

Report

Are this journal's acceptance criteria met?

No.

Recommend for another journal?

No.

Requested changes

NA

Attachment

---

## Editorial Decision

resubmitted